# Adsorption Characteristics of Polymer Solutions on Media Surfaces and Their Main Influencing Factors

**DOI:** 10.3390/polym13111774

**Published:** 2021-05-28

**Authors:** Shijie Zhu, Zhongbin Ye, Zhezhi Liu, Zhonghua Chen, Jun Li, Zuping Xiang

**Affiliations:** 1Institute of Petroleum and Natural Gas Engineering, Chongqing University of Science and Technology, Chongqing 401331, China; liuzhezhi1@163.com (Z.L.); 2014013@cqust.edu.cn (Z.C.); 2011927@cqust.edu.cn (J.L.); 2014006@cqust.edu.cn (Z.X.); 2Chengdu Technological University, Chengdu 610031, China; yezhongb@126.com; 3State Key Laboratory of Oil & Gas Reservoir and Exploitation Engineering, Southwest Petroleum University, Chengdu 610500, China

**Keywords:** partially hydrolyzed polyacrylamide, hydrophobically associating polymer, dendrimer, isotherm adsorption model, adsorption kinetic model

## Abstract

In practical applications, the chemical and physical adsorption of a polymer solution greatly affects its action mode and effect. Understanding the adsorption mechanism and its influencing factors can help to optimize the application mode and ensure application efficiency. Three types of polymer solutions—partially hydrolyzed polyacrylamide (HPAM), hydrophobically associating polymer (AP-P4), and dendrimer hydrophobically associating polymer (DHAP), which are viscoelastic liquids—were used as sorbates to study their adsorption by a sorbent such as quartz sand. The effects of the solution concentration, contact time, particle size of quartz sand, solid–liquid ratio, and fluid movement on the adsorption capacity of the polymer solutions were examined. The results showed that HPAM presents a typical Langmuir monolayer adsorption characteristic, and its adsorption capacity (per unit area) is 1.17–1.62 μg/cm^2^. The association enhances the interactions of the AP-P4 and DHAP solutions, and they present multilayer characteristics of first-order chemical adsorption and secondary physical molecule adsorption. Moreover, the dendrite structure further increases the adsorption thickness of DHAP. Hence, the adsorption thicknesses of AP-P4 and DHAP are four and six times that of HPAM, respectively. The adsorption of the three polymers is consistent with the influence of fluid motion and decreases with increasing fluid velocity. However, the larger the thickness of the adsorption layer, the clearer the influence of the flow, and the higher the decrease in adsorption capacity. Optimizing the injection rate is an effective method to control the applications of a polymer in porous media.

## 1. Introduction

Polymer flooding has been broadly applied as a major enhanced oil recovery (EOR) technology in numerous oilfields worldwide [1,2,3]. Polymer flooding technology increases the displacement phase viscosity and reduces the displacement phase permeability, thereby improving unfavorable mobility ratios, controlling the displacement front edge, and enhancing the displacement and sweep efficiencies [4,5,6]. The decrease in the displacement phase permeability is caused by the comprehensive outcomes of polymer solution adsorption and retention in porous media [7]. Rising adsorption capacity is accompanied by increasing adsorption retention capacity, the ability to reduce permeability, and the sweep ability of the subsequent fluid [8,9]. There are two main methods for the study of polymer solution adsorption: static and dynamic adsorption [10,11,12,13]. The determination of dynamic adsorption is based on the total amount of adsorption and retention, not on a single adsorption variable [14,15,16,17]. Although some static adsorption methods are unrelated to the actual adsorption state, they are easier to conduct and explain [18,19,20]. Therefore, it is more feasible to study adsorption characteristics and influencing factors based on static adsorption methods.

Currently, the most widely used polymer solutions for oil displacement in oilfields are partially hydrolyzed polyacrylamide (HPAM) and the hydrophobically associating polymer AP-P4 [21,22,23,24]. The use of dendrimer hydrophobically associating polymers (DHAPs) is one of the leading research methods to improve complex reservoir conditions [25,26]. Among them, the adsorption mechanism and influencing factors of HPAM are more prominent. Its mechanism is chemisorption on the surface of the medium by electrostatic and hydrogen bonding forces, forming a stable monolayer feature [27,28]. For hydrophobically associating polymers, the free polymer solution molecules are entangled with the adsorbed polymer molecules due to association, which exhibits a multimolecular layer with increasing adsorption [29,30]. However, there are only a few reports on the adsorption mechanism of dendrimers. Some scholars studied the adsorption law and found that the dendrimer/hyperdendrimer polymer static adsorption law conforms to the Langmuir adsorption characteristics and those of a multimolecular layer [31].

It is inaccurate to describe polymer adsorption only by the Langmuir adsorption characteristic definition, and the understanding and research of its adsorption mechanism need further exploration. Various adsorption mathematical models have been applied for fitting [32,33,34,35,36,37,38] (Table 1).

It is relevant to discuss the adsorption mechanism of a polymer solution by combining the isothermal adsorption model with the adsorption kinetic model. Furthermore, it is essential to consider the potential adsorption factors of a polymer solution in porous media. These mainly include the following three aspects [39,40,41,42]: (1) the influence of the external force caused by the fluid movement on its adsorption, (2) the influence of the effective contact adsorption time with the medium surface in the flow process, and (3) the effective adsorption area between the fluid and the medium, and the influence of the concentration change caused by the fluid adsorption, i.e., the influence of the effective adsorption concentration. The adsorption characteristics and influencing factors of HPAM, AP-P4, and DHAP were studied by comparative experiments to fully understand the adsorption mechanism and fundamentally comprehend the influence of adsorption on seepage characteristics.

## 2. Experiment

### 2.1. Experimental Materials and Equipments

Experimental polymers: The molecular weight of HPAM (Sichuan Guangya Polymer Chemical Co., Ltd, Nanchong, China) is 20 million, and its molecular formula is illustrated in Figure 1. The hydrophobically associating polymer AP-P4 ((Sichuan Guangya Polymer Chemical Co., Ltd, Nanchong, China) has a molecular weight of 18 million, solid content of 88%, and degree of hydrolysis of 23.6%. Its molecular formula is depicted in Figure 2. DHAP has a relative molecular weight of 6 million and a hydrophobic group content of 0.6 mol %. The specific synthesis steps have been reported in previous literature [11,25,26], and its molecular formula is shown in Figure 3.

Experimental water: Distilled water was used to prepare 3 mg/mL of experimental salt solution (NaCl was analytically pure).

The experimental temperature was 20 °C.

Experimental quartz sand: Acid-washed quartz sand (pH approximately 7) had neutral wetting and high roundness. The mesh size range of the quartz sand was 20–160. An oscillator (Shanghai Ruang Technology Co., Ltd, Shanghai, China) and a screen were used to separate quartz sands with various mesh sizes, and the particle size was also analyzed. The mesh numbers were 20–40, 40–60, 60–80, 80–100, 100–120, 120–160, and >160.

The experimental equipment included a 1- L beaker, IKA RW20 digital mechanical stirrer (IKA works GmbH & Co. Staufen, Germany), TU-1901 dual-beam UV-Vis spectrophotometer (Shanghai Huyueming Scientific Instrument Co., Ltd, Shanghai, China), h2-16k centrifuge (Hunan Kecheng Instrument Equipment Co., Ltd, Changsha, China), 100 mL volumetric flask, magnetic stirrer (IKA works GmbH & Co. Staufen, Germany), and an A-type electronic balance (acs-30 kg (±2 grams)).

### 2.2. Experimental Contents and Steps

#### 2.2.1. Static Adsorption Capacity

The key experimental parameters of the isotherm adsorption model were constructed based on the static adsorption experimental data. Different mathematical models were fitted based on the experimental data, and part of the adsorption mechanism was analyzed. The static adsorption experiment was conducted by the soaking method, using a UV spectrophotometer to determine the absorbance [11]. The specific steps were as follows: To prepare polymer solutions of different concentrations, 100 mL of polymer was placed in a wide-mouth bottle. Acid-washed quartz sand was added at a solid to liquid ratio of 1:9 with stirring, while the maximum contact time was 24 h. The supernatant was taken, and the concentration of the solution was diluted to 0.02–0.1 mg/mL. Then, each solution was mixed for 2 h, centrifuged on the first gear for 5 min, and kept to detect absorbance. The absorbance of each polymer solution was measured using a UV spectrophotometer, and the adsorption capacity was calculated using the standard curve data. The standard absorbance curves of HPAM, AP-P4, and DHAP are expressed by Equations (1)–(3).
HPAM: Abs = 0.024 × C(mg/mL) − 0.4409, R^2^ = 0.9907.(1)
DHAP: Abs = 0.0203 × C(mg/mL) − 0.3121, R^2^ = 0.9991.(2)
AP-P4: Abs = 0.0191 × C(mg/mL) − 0.3031, R^2^ = 0.9875.(3)

#### 2.2.2. Effect of Contact Time on Adsorption Capacity

The relationship between the adsorption amount and time can be established by studying the capacity of quartz sand to adsorb polymer from solution at different periods of contact. This yields the fitting parameters of the adsorption kinetic model and allows for analyzing the partial adsorption mechanism of the polymer solution. The specific experimental conditions were as follows: The concentration of polymer solution was 2 mg/mL, whereas 100 mL of this solution was contained in a wide-mouth bottle. Acid-washed quartz sand was added to polymer solution at a solid to liquid ratio of 1:9. The liquid system was shaken for a contact time from 1 to 1440 min.

#### 2.2.3. Influence of Solid to Liquid Ratio on Adsorption Capacity

The influence of different solid to liquid ratios on adsorption value was studied; ratios varied from 2:8 to 9:1.

#### 2.2.4. Impact of Effective Contact Area on Adsorption Capacity

The unit of polymer adsorption capacity can be generally expressed in μg/g or mg/g of sorbent. However, for comparative analysis, it is more useful to use the amount of polymer adsorption per unit surface area of quartz sand. The specific experimental conditions were as follows: To simplify the calculation of the surface area (Table 2), the shape of the selected fraction of sand particles was taken to be spherical. The concentration of polymer solution was 2 mg/mL, while the sand was kept entirely in contact with this solution for 12 h. The adsorption capacity was determined after separation of polymer solution and sand.

#### 2.2.5. Influence of Fluid Movement on Adsorption Capacity

The addition of quartz sand was optimized through adsorption tests of the sand and polymer solutions in the static adsorption experiment process. This was conducted to reduce the impact of the pore throat structure of porous media on the dynamic retention of the polymer solutions. Specifically, a small amount of quartz sand was placed on the bottom surface of a bottle without overlapping. The polymer solution was then added according to the solid–liquid ratio of 1:9 (two concentrations of 1 mg/mL and 2 mg/mL were selected). The flasks containing the mixtures were placed on a magnetic stirrer to control the rotation speeds at 30, 60, 120, 240, 480, 960, and 1920 rad/min. After full contact for 24 h, the supernatant was removed, and the polymer solution concentration was determined. The linear velocity was transformed by the rotating speed of the moving rotor, and the rotor diameter was taken as the boundary condition. The linear velocity was calculated by considering the rotor diameter as the rotation length; this formula is expressed in Equation (4).
υ = 2πR × ω(4)
where *v* is the linear velocity, m/s; ω is the angular velocity, rad/s; and R is the radius, m.

## 3. Results and Discussion

### 3.1. Static Adsorption Capacity

The static adsorption capacity results of the HPAM and DHAP solutions on the quartz sand surface are summarized in Table 3. The adsorption capacity of HPAM and DHAP increases with increasing solution concentration and tends to be stable. The static equilibrium adsorption capacities of polymers HPAM, AP-P4, and DHAP are approximately 200, 780, and 1100 μg/g, respectively.

The three isotherm adsorption models listed in Table 1 were used to fit the experimental data in Table 3. The Langmuir and Temkin isotherm adsorption model fitting results are better than those of the remaining models. The data are shown in Figure 4, Figure 5 and Figure 6.

The fitting results of the Langmuir isotherm adsorption model for the polymer solutions are presented in Figure 4. The Langmuir fit is extremely good for the three polymer solutions; however, the fitting degrees of AP-P4 and DHAP are lower than that of HPAM. The results of the Temkin isotherm adsorption model (Figure 5) confirm that only HPAM has a high fitting degree (89.12%). Based on the fitting formula and results, the adsorption of HPAM is consistent with the results obtained by various scholars. Polymer HPAM is adsorbed uniformly on the quartz sand surface, which is chemisorbed with strong intermolecular forces. There is no interaction between the adsorbate molecules, comparable to single molecular layer adsorption characteristics, with the maximum equilibrium adsorption capacity. The free polymer molecules are considered entangled with the adsorbed polymer molecules via hydrophobic association, raising the adsorption capacity. In contrast, the fitting accuracy of the mathematical model is reduced, particularly for the Temkin model.

### 3.2. Effect of Contact Time on the Adsorption Capacity Polymers

The adsorption experiment outcomes of polymers HPAM, DHAP, and quartz sand under various contact times are summarized in Table 4. As the polymer solution and quartz sand contact period increases, the polymer solution adsorption capacity gradually increases and tends to stabilize after a specific time [43,44,45]. The equilibrium times of HPAM, AP-P4, and DHAP adsorption are 45 min, 6 h, and 6 h, respectively.

As listed in Table 1, the four model fitting degrees are higher than those of the pseudo-second-order kinetic and Elovich models. The experimental results are shown in Figure 7, Figure 8, Figure 9 and Figure 10.

The fitting results of the four kinetic models were compared. The pseudo-second-order kinetic model fitting of the polymer solutions is very high, indicating that chemisorption is the primary polymer adsorption mechanism on the quartz sand surface. In the fitting of the Elovich model, DHAP and AP-P4 have high fitting degrees; specifically, the adsorption rate of DHAP, which decreases exponentially with increasing adsorption capacity. This finding explains the multimolecular layer adsorption and the extended time required to achieve adsorption equilibrium. The polymer solution contact adsorption with quartz sand mainly occurs by electrostatic and hydrogen bond chemisorption, forming a relatively stable monolayer adsorption layer. Subsequently, the entanglement between the free and adsorbed polymer molecules leads to the above-mentioned multimolecular layer formation. The winding process formed by the association is also a winding/unwinding dynamic equilibrium process. Thus, the adsorption equilibrium time of the polymer is significantly prolonged. The fitting degree of HPAM is low, implying the adsorption rate does not decrease exponentially with increasing adsorption amount. Chemisorption is the only adsorption observed, and only single molecular layer equilibrium adsorption occurs.

In the quasi-second-order kinetics, the adsorption rate index of HPAM is the highest, and its single chemical adsorption mechanism helps it reach adsorption equilibrium as soon as possible. The multimolecular layer entangled by association reduces the adsorption rate. The interaction force changes the relationship between free and adsorbed molecules. The DHAP dendrimer interaction is strong, so its adsorption rate index is the lowest. The fitting of the kinetic model establishes the adsorption mechanism and demonstrates the inferior ability of hydrophobically associating polymers and dendrimers to achieve stable adsorption. However, the adsorption capacity is large. The quasi-second-order kinetic and Elovich model high-precision fitting results further illustrate the importance of considering effective solution concentration, effective contact area, and flow movement on the adsorption capacity.

### 3.3. Influence of Effective Concentration on Adsorption Capacity

The experimental outcomes with the varying solid–liquid ratios are exhibited in Figure 11.

The liquid content slowly decreases with increasing solid content in the solid–liquid mixture, and the polymer solution adsorption capacity decreases. When the solid–liquid ratio is 3:7, the equilibrium adsorption capacity of the three polymers decreases when the liquid content is further reduced. Changes in the solid–liquid ratio lead to variations in the relative concentration of solution adsorption. However, for polymer adsorption in porous media, the effect of this phenomenon is not clear. Because the flowing polymer solution continuously pushes the high-concentration polymer solution forward, the medium surface at the current position is continuously in contact with the high-concentration polymer solution to achieve adsorption equilibrium. Due to the decrease in the effective polymer concentration at the displacement front, the adsorption characteristics vary, and the time to reach adsorption equilibrium is prolonged.

### 3.4. Impact of Effective Contact Area on Adsorption Capacity

The influence of the quartz sand particle size on the polymer solution adsorption capacity is presented in Figure 12.

The static adsorption of the three polymers increases with the increase in the quartz sand particle size. The results confirm that the total surface area of the quartz sand particles (with identical quality) increases with particle size reduction. More adsorption surfaces are accessible to the polymers, and the multilayer adsorption characteristics expand, raising the adsorption quantity.

Based on the adsorption capacity achieved with different particle sizes, the surface area adsorption capacity of the three polymers can be obtained by conversion, as summarized in Table 5.

The adsorption capacities of HPAM and DHAP are 1.17–1.62 and 6.44–8.45 μg/cm^2^, respectively. HPAM is characterized by monolayer adsorption (see the adsorption diagram in Figure 13), and the unit surface area adsorption of polymer AP-P4 is 4.26–6.67 μg/cm^2^. The adsorption of DHAP per unit area is almost six times that of HPAM, which indicates that its adsorption thickness is six times that of HPAM. This can be attributed to the dendrite structure providing the molecules with a more extensive spatial distribution, increasing the molecular layer thickness, and leading to more noticeable multimolecular layer adsorption.

### 3.5. Influence of Fluid Movement on Adsorption Capacity

The dynamic adsorption capacities of polymer solutions with concentrations of 1 and 2 mg/mL were determined. The experimental results are summarized in Table 6.

Based on static experiments, the movement of the polymer solution has a specific impact on its quartz sand adsorption. As the solution velocity increases, its equilibrium adsorption capacity in porous media gradually declines, suggesting that the movement affects the adsorption of the polymer on the quartz sand surface. Moreover, the higher the movement speed, the larger the pull adsorption effect of the polymer, and the more enhanced the desorption effect of the adsorbed polymer under an external force. Different polymer solutions are affected by different velocities. Single chemisorption of HPAM is the least affected, followed by hydrophobically associating polymer AP-P4, with DHAP being the most affected. Among the polymers, DHAP has the least number of adsorption sites and the lowest adsorption force on the quartz sand surface, and the external force of the fluid has a strong influence.

By comparing the ratio of dynamic and static adsorption capacity under different stirring rates (see Figure 14), the following were determined: There is no association between HPAM and concentrations of 1 and 2 mg/mL. The increase in solution concentration raises the intermolecular interaction force, enhances the adsorption force on the surface of the medium, and decreases under the influence of external force. Under the influence of hydrophobic association, the entanglement force between AP-P4 and DHAP molecules increases considerably, which is affected by the external force. With an increasing stirring rate, the entanglement force rapidly drops and tends to stabilize, indicating that the physical adsorption of the multimolecular layer formed by association is a reversible process. Before the critical association concentration, the hydrophobic group of the molecular chain produces part of the intramolecular association, which increases the intramolecular force to a certain extent and is more affected by the external force. After the critical association concentration, the intermolecular association dramatically increases the interaction between free and adsorbed molecules, so the external force has a more prominent effect. Compared with the concentration of 1 mg/mL, the adsorption capacity of 2 mg/mL solution decreases with an increased stirring rate. The branching structure of the polymer solution cannot change its adsorption force on the surface of the medium, it can only increase the thickness of the adsorption layer through its stretching spatial structure. The key to the balance between adsorption and external force is the main factor affecting the dynamic adsorption capacity. Changing the flow state of the polymer fluid in porous media can improve its adsorption capacity, laying the foundation for the technical guidance of changing mobility control through technological means.

## 4. Conclusions

The equilibrium adsorption of HPAM, AP-P4, and DHAP from solutions by quartz sand at optimal conditions are 200, 780, and 1100μg/g, respectively.The monomolecular adsorption of HPAM from its solutions on the sand surface is in good agreement with the Langmuir and quasi-second-order kinetic model.The adsorption of AP-P4 and DHAP polymers from solutions on the sand surface is multimolecular. Since the dendritic structure of polymers increases the thickness of the adsorption layer, the thickness of the adsorption layer of AP-P4 and DHAP is four and six times that of HPAM, respectively.To achieve maximum adsorption value, the solid–liquid ratio should be less than 3:7. The larger the effective adsorption area, the greater the adsorption capacity of the polymer solution. Additionally, the larger the fluid movement, the more apparent the decrease in the adsorption capacity. The greater the thickness of the adsorption layer, the more evident the fluid movement.Changing the flow state of polymer fluid in porous media can improve its adsorption capacity, laying the foundation for the technical guidance of changing mobility control through technological means.

## Figures and Tables

**Figure 1 polymers-13-01774-f001:**
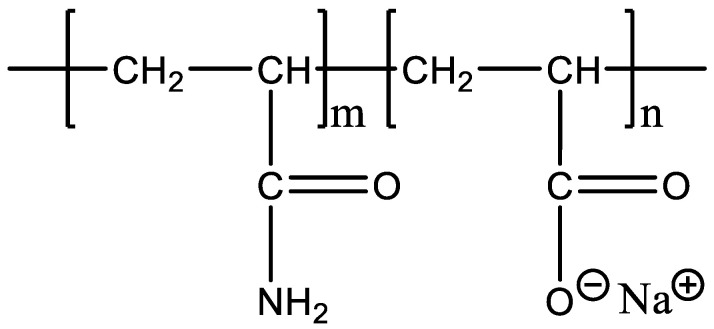
The molecular formula of HPAM.

**Figure 2 polymers-13-01774-f002:**
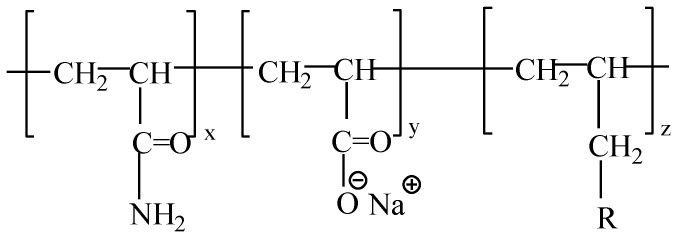
The molecular formula of AP-P4.

**Figure 3 polymers-13-01774-f003:**
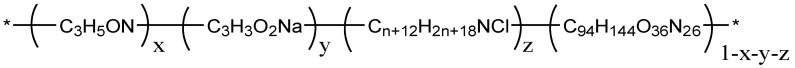
The molecular formula of DHAP.

**Figure 4 polymers-13-01774-f004:**
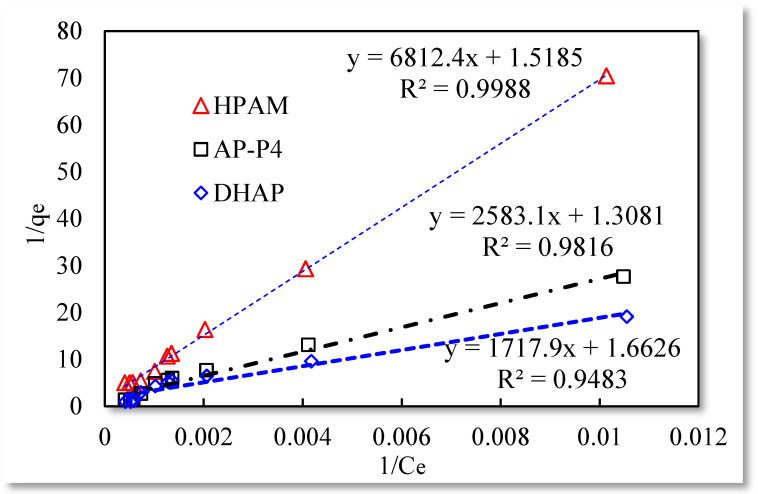
Langmuir isotherm adsorption model.

**Figure 5 polymers-13-01774-f005:**
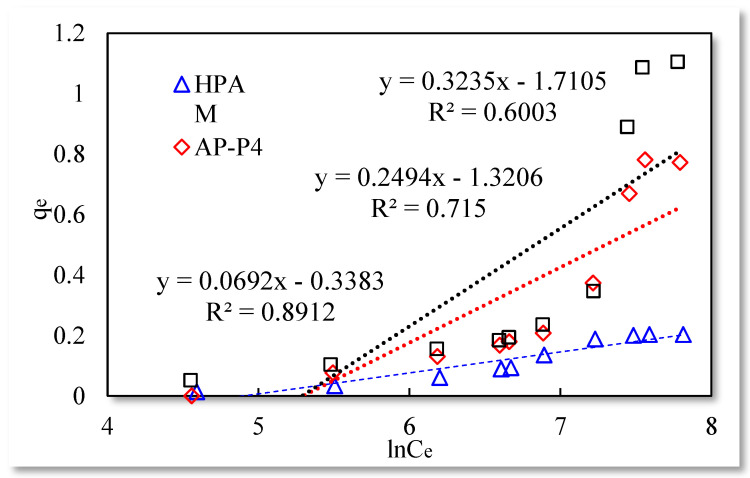
Temkin isothermal adsorption model.

**Figure 6 polymers-13-01774-f006:**
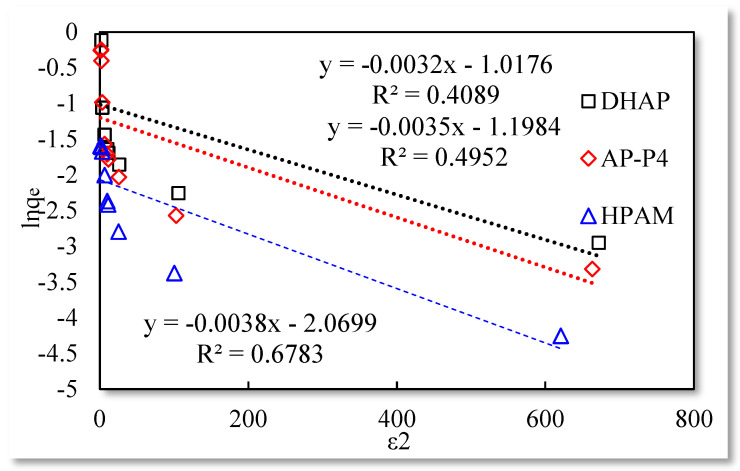
Dubinin−Radushkevich isothermal adsorption model.

**Figure 7 polymers-13-01774-f007:**
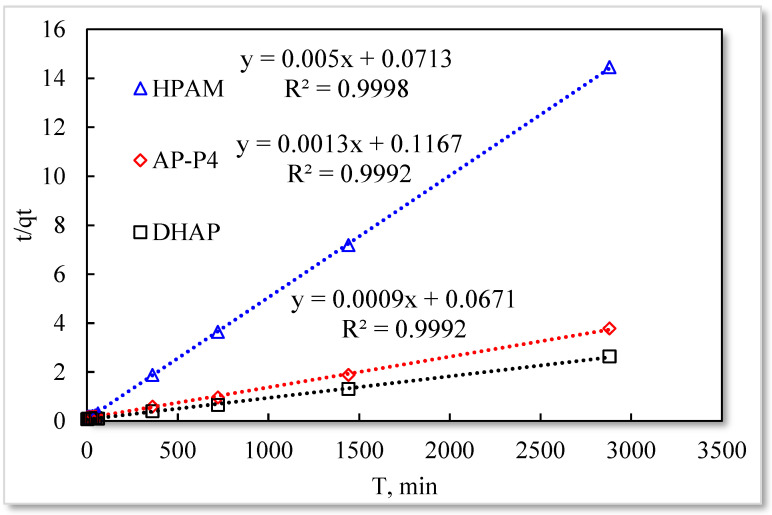
Pseudo-second-order kinetic model.

**Figure 8 polymers-13-01774-f008:**
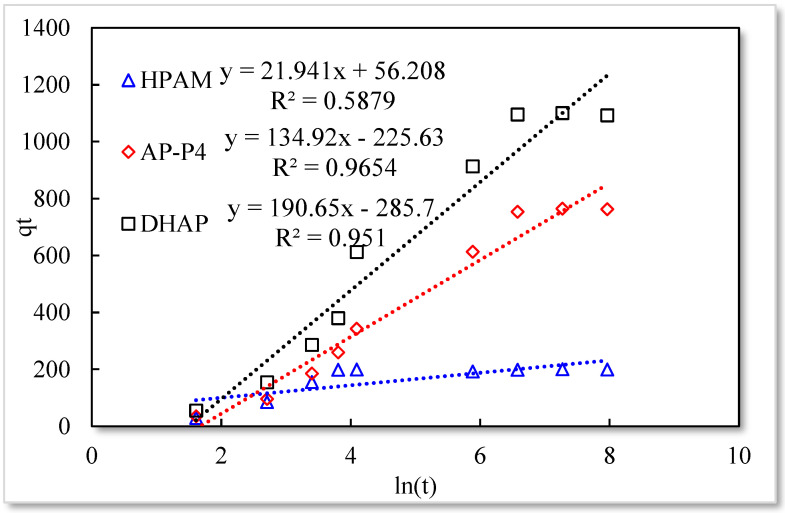
Elovich model.

**Figure 9 polymers-13-01774-f009:**
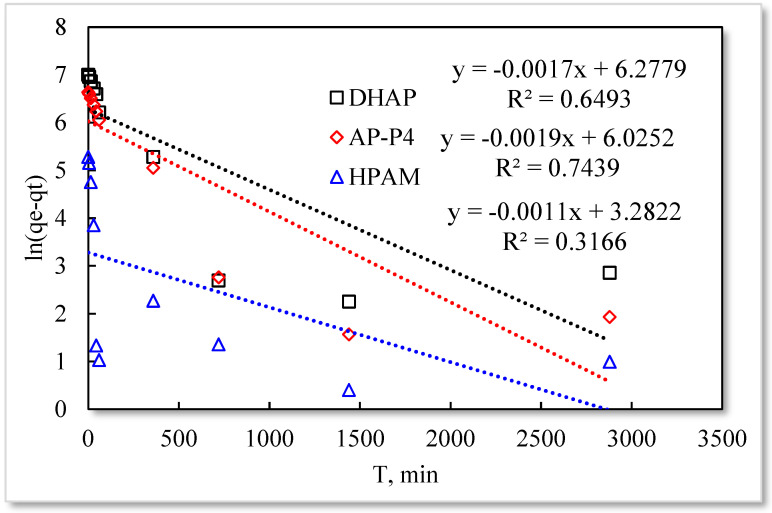
Pseudo−first−order kinetic model.

**Figure 10 polymers-13-01774-f010:**
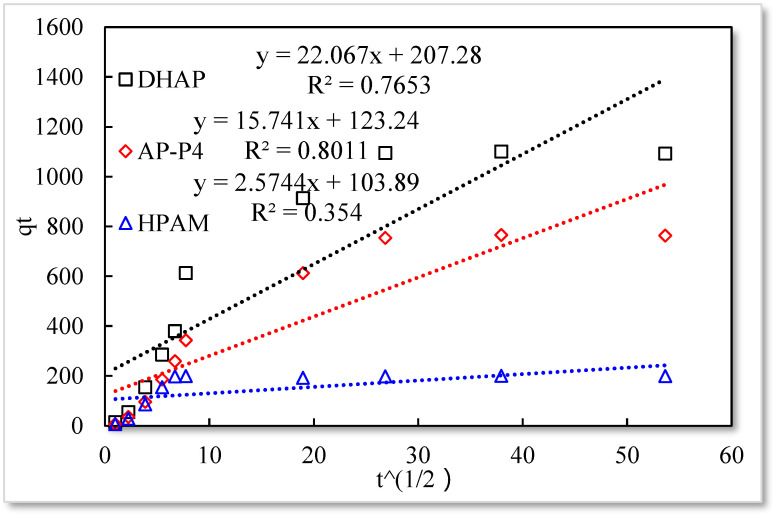
Particle diffusion model.

**Figure 11 polymers-13-01774-f011:**
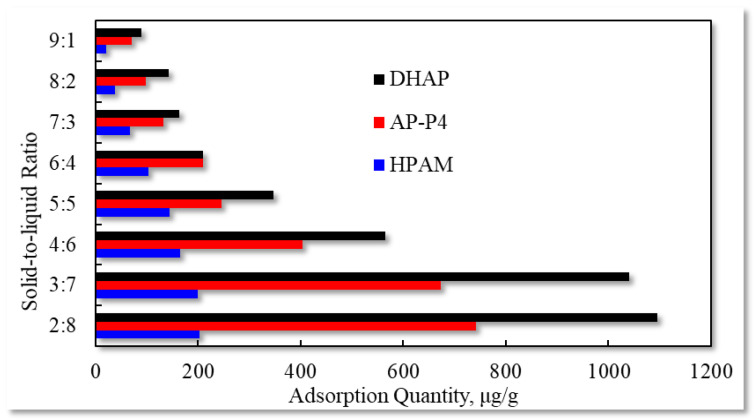
Adsorption capacity characteristics for different solid–liquid ratios.

**Figure 12 polymers-13-01774-f012:**
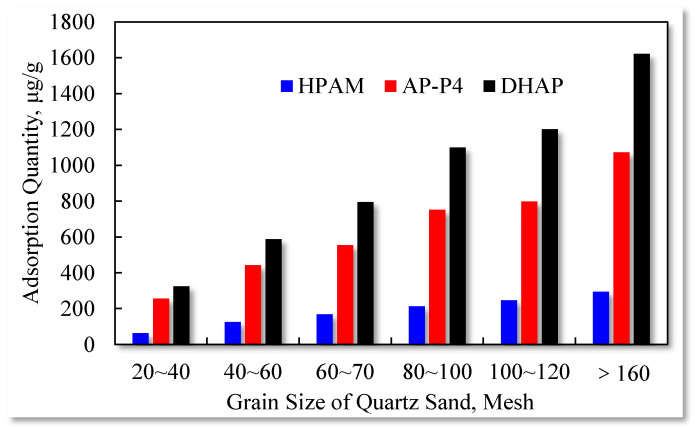
Effect of quartz sand particle size on polymer adsorption capacity.

**Figure 13 polymers-13-01774-f013:**
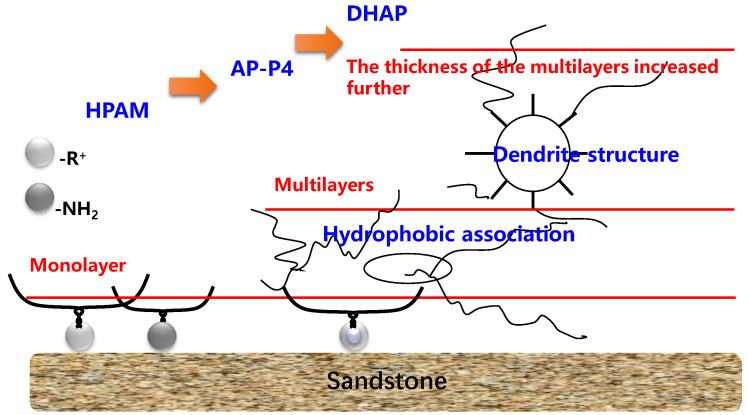
Adsorption characteristics of the polymer.

**Figure 14 polymers-13-01774-f014:**
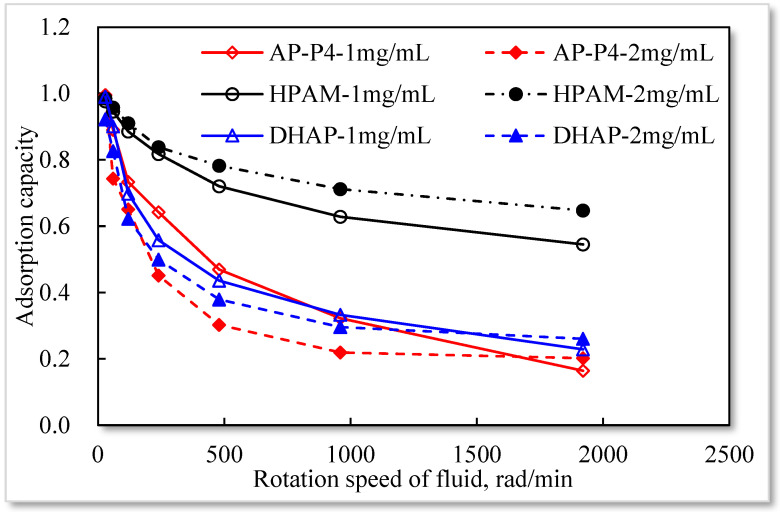
Ratio analysis of dynamic to static adsorption capacity under different stirring rates.

**Table 1 polymers-13-01774-t001:** Adsorption model formula and its physical significance.

Category	Model and Formula	Physical Meaning of Formula
Isothermal adsorption model	Linear expression of Langmuir model $\frac{1}{q_{e}}=\frac{1}{q_{m}}+\frac{1}{k_{L}q_{m}C_{e}}$	The model assumes that only one adsorbate molecule is adsorbed on each activator site. The attachment sites are identical and undifferentiated. The adsorption force is strong enough and belongs to chemical or physical force. The adsorbate does not move on the surface of adsorbent. The adsorption capacity is maximized when the adsorbate is saturated on the surface of adsorbent.
	Linear expression of Temkin $q_{e}=Blnk_{T}+BlnC_{e}$	The model is suitable for adsorbents with heterogeneous surfaces and is often used to describe adsorption processes in which there are strong intermolecular interactions between adsorbate and adsorbent, such as strong electrostatic interaction or ion exchange interaction
	Dubinin–Radushkevich model $lnq_{e}=lnq_{m}-\beta \varepsilon^{2}$	The model assumes that the adsorption energy is heterogeneous and the distribution of adsorption energy is Gaussian
Adsorption kinetic model	Pseudo-second order adsorption kinetics model $\frac{t}{q_{t}}=\frac{1}{k_{2}q_{e}^{2}}+\frac{t}{q_{e}}$	The model assumes that the adsorption rate is directly proportional to the square of the adsorbate concentration, and the limiting factor of the adsorption rate is the adsorption mechanism. Chemisorption is the only or the most essential adsorption mechanism, and the adsorption reaction occurs by sharing or gaining/losing electrons between the adsorbent and adsorbate.
	Elovich model $q_{t}=a+blnt$	The model assumes that the adsorption energy is not uniform and increases linearly with the increase in surface coverage. The adsorption rate is not uniform, but decreases exponentially with the increase in adsorption capacity.
	Particle diffusion model $q_{t}=k_{ip}t^{1/2}+C_{i}$	The model assumes that the driving force of the adsorption process is from the concentration gradient of the adsorbate in the solution.
	Pseudo-first order adsorption kinetic model $ln\left(q_{e}-q_{t}\right)=lnq_{e}-k_{1}t$	The model assumes that the adsorption rate is directly proportional to the concentration of adsorbate, and the factor limiting the adsorption rate is the mass transfer resistance in the particle.

**Table 2 polymers-13-01774-t002:** Surface area, cm^3^, of 1 g quartz sand with varying mesh sizes.

Mesh	20–40	40–60	60–70	80–100	100–120	>160
Radius, cm	0.0295	0.015	0.011	0.008	0.007	0.0045
Surface area, cm^3^, of 1 g quartz sand	38.375	75.472	102.916	141.510	161.725	251.572

**Table 3 polymers-13-01774-t003:** Static adsorption capacity at various concentrations.

Concentration (mg/mL)	0.1	0.25	0.5	0.75	0.8	1	1.4	1.8	2	2.5
HPAM, μg/g	14.6	33.6	70.7	87.2	92.4	132.4	189.2	197.3	201.2	201.1
AP-P4, μg/g	42.3	84.1	121.8	157.3	170.0	209.1	502.5	642.7	764.2	784.2
DHAP, μg/g	52.4	104.6	156.2	184.3	194.2	236.8	347.2	890.1	1087.3	1105.7

**Table 4 polymers-13-01774-t004:** Effect of contact time on polymer adsorption capacity.

Contact Time (min)	1	5	15	30	45	60	360	720	1440	2880
HPAM, μg/g	5.9	28.6	85.4	154.8	198.2	199.2	192.3	198.1	200.5	199.3
AP-P4, μg/g	6.9	35.2	96.3	185.3	259.6	342.3	613.1	754.2	765.2	763.1
DHAP, μg/g	13.9	55.2	154.3	285.3	379.6	612.3	913.1	1095.2	1100.5	1092.6

**Table 5 polymers-13-01774-t005:** Adsorption capacity of various fractions of quartz sand particles.

Mesh	20–40	40–60	60–70	80~100	100–120	>160
HPAM, μg/cm^2^	1.62	1.66	1.63	1.51	1.52	1.17
AP-P4, μg/cm^2^	6.67	5.86	5.38	5.31	4.93	4.26
DHAP, μg/cm^2^	8.45	7.78	7.73	7.76	7.43	6.44

**Table 6 polymers-13-01774-t006:** Effect of movement speed on polymer adsorption capacity.

Polymer	Adsorption Capacity, μg/g	Rotation Speed of Fluid, rad/min						
		30	60	120	240	480	960	1920
	Concentration, mg/mL	Linear Speed Corresponding to Different Rotation Speed, m/s						
		0.01	0.02	0.04	0.08	0.16	0.32	0.64
HPAM	1	129.3	125.2	117.2	108.2	95.4	83.2	72.1
	2	198.2	192.6	183.2	168.6	157.3	143.2	130.2
AP-P4	1	206	186.2	153.2	134.2	98.2	67.3	34.3
	2	760.4	567.8	497.2	345.1	231	167.8	154.2
DHAP	1	234.5	213.2	165.2	132.1	103.2	78.7	54.2
	2	1002.3	898.2	676.3	543.2	412.3	321.3	283.2

## Data Availability

The data supporting the findings of this study are available from the corresponding author upon reasonable request.

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
