# Peer review of "Adsorption Characteristics of Polymer Solutions on Media Surfaces and Their Main Influencing Factors"

_polymers, 2021, doi:10.3390/polym13111774_

Round 1

Reviewer 1 Report

The paper is well written. I only have one concern.  That's regarding the suggestion that a monolayer is detected.  Often, authors seem to jump to this conclusion without realizing that the data don't quite support that conclusion.  I certainly see several graphs, where there is significant deviation from the Langmuir monolayer model.

Author Response

Thank you very much for your approval. In fact, monolayer membrane is based on a consensus understanding of polymer HPAM. Many literatures at home and abroad have proved the content of monolayer adsorption, so the author did not use any means to prove it in the process of research. The adsorption of monolayer is to compare with that of multilayer, so some of the language formulas are deviated.

Reviewer 2 Report

Lines 93-94. Experimental water. Remark: “Experimental water” should be replaced with “Salt solution”. Since term “saline” is used in medicine and biology as NaCl solution with concentration of 9 mg/mL (0.9%), the “saline” should be replaced with “salt solution”. The concentration of the salt solution is best expressed shortly in mg/mL, i.e. 3 mg/mL, instead of 3000 mg/L. Thus, this sentence should be edited, e.g. as follows: “Distilled water was used to prepare a salt solution with concentration of 3 mg/mL (NaCl was analytically pure). In addition, it is necessary to clarify where this salt solution was used in this paper and for what purpose?

Lines 112-113. 1). To prepare polymer solutions of different concentrations, 100 mL of polymer was placed in a wide-mouth bottle. Remark: 100 mL is no polymer, but polymer solution. Therefore, this sentence should be clarified, e.g. “To prepare polymer solutions of different concentrations, 100 mL of polymer solution with concentration of  2 mg/mL (?) was placed in a wide-mouth bottle”.

Lines 113-117. Remark: This section must be corrected, e.g. Acid-washed quartz sand was added at a solid to liquid ratio of 1:9 with stirring to ensure the solution could be stirred and mixed, while maximum contact time was 24 h; 2) The supernatant was taken, and the concentration of the solution was diluted to 0.02–0.1 mg/mL.  Then, each solution was mixed for 2 h, centrifuged on the first gear for 5 min, and kept to detect absorbance;

Line 119. The standard absorbance curves of HPAM, AP-P4, and DHAP are expressed in equations. Remark: This sentence should be clarified and corrected: “The standard absorbance curves of HPAM, AP-P4, and DHAP solutions are expressed by equations”

Equations (1), (2), (3). Remark: Units of concentration C must be specified.

 Lines 125-127. The relationship between the adsorption amount and time can be established by studying the adsorption capacity of a polymer solution and quartz sand under different periods of contact. Remark: Since the adsorption capacity refers to sorbent, this sentence should be corrected, e.g. “The relationship between the adsorption amount and time can be established by studying the capacity of quartz sand to adsorb polymer from solution at different periods of contact”.

Lines 128-132.  Remark: This section must be corrected, e.g. “The specific experimental steps conditions were as follows. The concentration of polymer solution was 2 mg/mL, whereas 100 mL of this solution was were contained in a wide-mouth bottle. Acid-washed quartz sand was added to polymer solution at a solid to liquid ratio of 1:9. The mixed liquid system was shaken for a contact time from 1 to 1440 min.

Lines 134-136.  Remark: This section must be corrected, e.g.

2.2.3. Influence of solid to liquid ratio on adsorption capacity

“The influence of different solid to liquid ratio on adsorption value was studied, taken this ratio from 2:8 to 9:1.

Line 138-146. Remark: This section must be corrected, e.g.

“The unit of polymer adsorption capacity for oil displacement can be generally expressed in μg/g or mg/g of sorbent. However, for comparative analysis,  it is more useful to use the amount of polymer adsorption per unit surface area of quartz sand. The specific experimental conditions were as follows. To simplify the calculation of the surface area (Table 2), the shape of the selected fraction of sand particles was taken to be spherical.

By optimizing the quartz sand particles with enhanced roundness (to 141 simplify the calculation of the surface areas of spherical particles, see Table 2), particle 142 sizes of 20–40, 40–60, 60–80, 80–100, 100–120, and >160 meshes were selected.

For the same mass of quartz sand

The concentration of polymer solution was 2 mg/mL, while the sand was kept entirely in contact with this solution for 12 h. The adsorption capacity was determined after separation of polymer solution and sand.

Table 2. Remark: (1). The unit of surface area of 1 g of quartz sand is missed; (2). In any case, values of this surface area (S) were calculated incorrectly. The correct eq. to calculate S of round (spherical) particles is the following:

S (cm2/g) =3/(R d)

where R is average radius, cm; d=2.65 g/cm2 is specific gravity of quartz.

Example 1. If mesh 20-40 and R=0.0295 cm, then S=38.375 cm2/g or 0.00384 m2/g instead of 0.007421 in Table 2.

Example 2. If mesh >160 and R=0.0045 cm, then S=251.57 cm2/g or 0.0251 m2/g instead of 0.001132 in Table 2.

Since results of S in Table 2 are incorrect, all these results must be recalculated.

Result section also needs careful editing.

General remarks:

a). Regarding polymer solutions. What solvent was used to prepare polymer solutions? Was it water or another solvent? In addition, the manuscript does not contain a procedure for preparing polymer solutions, and this omission should be corrected. In addition, it is recommended to express the concentration of polymer & salt solutions in mg/mL instead of mg/L.

b). Regarding adsorption of polymers. If we are talking about adsorption, then there must be sorbate and sorbent. It is clear, that sorbate is polymer, but unclear that sorbent was used? Was it a sand? If yes, this sorbent must be indicated in abstract and in text of manuscript. For example in abstract: “Three types of polymer solutions—partially hydrolyzed polyacrylamide (HPAM), hydrophobically associating polymer (AP-P4), and dendrimer hydrophobically associating polymer (DHAP) that are viscoelastic liquids were used as sorbates to study their adsorption by a sorbent such as quartz sand”. Etc.

c). This paper is written carelessly. It contains many missing words and mistakes, making it difficult to understand the meaning. The level of English language is low, so the manuscript needs careful editing. In addition, scientific editing is also required.

 Thus, the manuscript of article cannot be recommended for publication in the presented form.

Round 2

Reviewer 2 Report

The authors have revised the manuscript in accordance with comments of the reviewer. However, even the revised manuscript contains still a number of shortcomings that need to be corrected.

In Table 2, the unit of surface area, cm3, must be added, i.e. “Surface area, cm3, of 1 g quartz sand

Table 5. Adsorption capacity per unit area of polymer  (?). Remarks: (1). The title is wrong because unit area relates to sorbent – quartz sand, and not to polymer. (2). The title of this table does not does not match its content, because mesh value is used instead of surface area of quartz sand particles.  Thus, the title should be rewritten as e.g.: Adsorption capacity of various fractions of quartz sand particles. (3). Sentence “Area adsorption capacity –“  must be removed from this table as unnecessary. (4). From the data in Table 2, it follows that with an increase in the mesh values, the specific surface area of the sand particles increases and, accordingly, the adsorption capacity should also increase. However, in Table 5 an opposite tendency is observed, which is wrong.

Thus, the Table 5 needs to be completely revised: (a) The title should be rewritten, (b) Unnecessary sentence “Area adsorption capacity –“  must be removed, and (c) The adsorption capacity should  increase with increasing the mesh values.

Lines 265-268 and throughout the text: “adsorption capacity”. Remark: This term relates to sorbent and not to polymer. For sorbed polymers, term “adsorption capacity” is not used, instead the term “adsorption” (without capacity), or “adsorption value” should be used.

Conclusion should be revised.

  1. The equilibrium adsorption capacities of HPAM, AP-P4, and DHAP were 200 μg/g, 780 μg/g, and 1100 μg/g, respectively. Remark: (a) Word capacities should be removed because it relates to sorbent – quartz sand, and not to polymer. (b). Since the equilibrium adsorption value of polymer depends on conditions then “optimal conditions” should be added. Thus, write this conclusion, as follows: “The equilibrium adsorption of HPAM, AP-P4, and DHAP from solutions by quartz sand at optimal conditions were 200 μg/g, 780 μg/g, and 1100 μg/g, respectively.
  2. Polymer HPAM matches the Langmuir and quasi-second-order kinetic model well, which is characterized as chemisorption in a single molecular layer. Remark: The wording of this conclusion needs to be improved. In addition, the quartz sand does not contain reactive functional groups and therefore the conclusion about chemisorption is unreliable. I recommend correct the conclusion 2, e.g. as follows: “The monomolecular adsorption of HPAM from its solutions on the sand surface is in good agreement with the Langmuir and quasi-second order kinetic model”.

  1. Remark: The conclusion 3 must be edited, e.g. as follows: “The adsorption of AP-P4 and DHAP polymers from solutions on the sand surface is multimolecular. Since the dendritic structure of polymers increases the thickness of the adsorption layer, the thickness of the adsorption layer of AP-P4 and DHAP is four and six times that of HPAM, respectively.
  2. Remark: The first sentence in conclusion 4 must be edited, as follows: “To achieve maximum adsorption value the solid–liquid ratio should be less than 3:7”.

Decision: The noted shortcomings must be corrected. Only after additional correction this article can be recommended for publication.
